# New Actors Driving the Epithelial–Mesenchymal Transition in Cancer: The Role of Leptin

**DOI:** 10.3390/biom10121676

**Published:** 2020-12-15

**Authors:** Monserrat Olea-Flores, Juan C. Juárez-Cruz, Miriam D. Zuñiga-Eulogio, Erika Acosta, Eduardo García-Rodríguez, Ana E. Zacapala-Gomez, Miguel A. Mendoza-Catalán, Julio Ortiz-Ortiz, Carlos Ortuño-Pineda, Napoleón Navarro-Tito

**Affiliations:** 1Laboratorio de Biología Celular del Cáncer, Facultad de Ciencias Químico Biológicas, Universidad Autónoma de Guerrero, Chilpancingo 39090, Mexico; monseolea@uagro.mx (M.O.-F.); jcjuarezc91@gmail.com (J.C.J.-C.); miriamzuniga@uagro.mx (M.D.Z.-E.); Erk_Acosta@outlook.com (E.A.); rgeduardo24@gmail.com (E.G.-R.); 2Laboratorio de Biomedicina Molecular, Facultad de Ciencias Químico Biológicas, Universidad Autónoma de Guerrero, Chilpancingo 39090, Mexico; zak_ana@yahoo.com.mx (A.E.Z.-G.); mamendoza@uagro.mx (M.A.M.-C.); julioortiz@uagro.mx (J.O.-O.); 3Laboratorio de Ácidos Nucleicos y Proteinas, Facultad de Ciencias Químico Biológicas, Universidad Autónoma de Guerrero, Chilpancingo 39090, Mexico; carlos2pineda@hotmail.com

**Keywords:** leptin, signaling pathways, EMT, epithelial markers, mesenchymal markers, cancer

## Abstract

Leptin is a hormone secreted mainly by adipocytes; physiologically, it participates in the control of appetite and energy expenditure. However, it has also been linked to tumor progression in different epithelial cancers. In this review, we describe the effect of leptin on epithelial–mesenchymal transition (EMT) markers in different study models, including in vitro, in vivo, and patient studies and in various types of cancer, including breast, prostate, lung, and ovarian cancer. The different studies report that leptin promotes the expression of mesenchymal markers and a decrease in epithelial markers, in addition to promoting EMT-related processes such as cell migration and invasion and poor prognosis in patients with cancer. Finally, we report that leptin has the greatest biological relevance in EMT and tumor progression in breast, lung, prostate, esophageal, and ovarian cancer. This relationship could be due to the key role played by the enriched tumor microenvironment in adipose tissue. Together, these findings demonstrate that leptin is a key biomolecule that drives EMT and metastasis in cancer.

## 1. Introduction

The term epithelial–mesenchymal transition (EMT) was described in 1995 by Elizabeth D. Hay [1]. EMT is a reversible cellular process, in which the epithelial cells transdifferentiate to mesenchymal cells [2]; this process is characterized by the loss of epithelial markers including apico-basolateral polarity and cell–cell junctions and the acquisition of mesenchymal properties such as the ability to migrate and invade surrounding tissue [3]. EMT is carried out in three different biological contexts: (1) Embryonic development, essential for the differentiation and specialization of several cell types during embryonic development, (2) stem cell differentiation or tissue regeneration, associated with the reconstruction of damaged tissue or organs, and (3) tumorigenesis, associated with resistance to apoptosis and anoikis as well as invasion and metastasis [4,5,6]. Numerous actors involved in the activation of the EMT program associated with tumor progression have been described such as growth factors (TGF-β (transforming growth factor-β) and EGF (epidermal growth factor)) [7,8,9,10,11,12,13,14,15,16], cytokines (IL-6, IL-8 and TNF-α) [17,18], chemicals (cadmium, Nickel(II) chloride, Nickel(II) sulfate, and sodium arsenite) [19,20], fatty acids (palmitate, oleic acid and arachidonic acid) [21,22], hormones (estradiol, progesterone, L-thyroxine (T4), 3,5,3′-triiodo-L-thyronine (T3), and resistin) [23,24,25], and others (oxidative stress, hypoxia, and nicotine) [26,27]. 

Leptin is a hormone that, even though it primarily regulates food intake and energy expenditure, has been associated with cancer progression in recent years [28,29]. Leptin (from the Greek *leptos*) was discovered in 1994 by Zhang et al. [30,31], and is secreted mainly by adipocytes; however, it can also be secreted by tumor tissue [30,32]. In addition, other biological functions have been described for leptin such as regulation of the immune system [33] and reproductive system [34], or the induction tumor [35]. Leptin promotes tumor progression, by inducing signaling pathways that regulate cell migration and invasion [36,37] and the maintenance of cancer stem cells [38,39] and metastasis [35]. Interestingly, leptin has recently been described as an important player in promoting EMT [40]. 

Several studies carried out over the past 25 years leave us with the general understanding that leptin is a protein that contributes to pathological states. In in vitro, in vivo, and patient studies have demonstrated that leptin is an actor in the induction of EMT during tumor progression; however, the question remains: What is the mechanism for the activation of this process? Researchers around the world are faced with this complex question that, if answered one day, could pave the way for future research that focuses on the search for a new and perhaps more effective treatment for cancer.

The goals of this review are as follows: (1) To update the markers associated with the epithelial and mesenchymal phenotype, (2) to describe the inducers associated with EMT and tumor progression, (3) to discuss the signaling pathways induced by leptin, and (4) to report the most recent findings from in vitro, in vivo, and patient studies of the role of leptin as an inducer of EMT in cancer.

## 2. Epithelial–Mesenchymal Transition

In cancer, the epithelial–mesenchymal transition (EMT) is a reversible cellular reprogramming process that allows cancer cells with an epithelial phenotype to acquire mesenchymal characteristics [41]. EMT is critical for tumor malignancy, since cancer mesenchymal cells acquire migratory and invasive capacities, resistance to anoikis, immune evasion, stemness characteristics, chemoresistance and metabolic reprogramming, all of which lead to metastasis [42,43]. Even though EMT appears to be a strict complete process from epithelial cells to mesenchymal cells, intermediate states of EMT have been described; these are also known as partial, incomplete, or hybrid phenotype states [41]. These intermediate states reflect the cellular plasticity of EMT as cells adopt different identities along the epithelial–mesenchymal spectrum, transiently co-expressing both epithelial and mesenchymal markers and, thus contributing to intratumoral heterogeneity (Figure 1) [44,45].

### 2.1. Epithelial Markers

The loss of apicobasal polarity is one of the first steps observed during EMT. Several of the mediators of cell junctions have been characterized as classic epithelial markers, such as E-cadherin in the adherent junctions; claudins, occludins, and ZO proteins in tight junctions; and the loss of desmoplakin in desmosomes [46]. However, there is also the reorganization of the cytoskeleton in which the loss of epithelial cytokeratins such as 8, 18, and 19 has been described canonically [47].

Although little is known about the transcription factors that maintain the epithelial phenotype, it is known that the transcription factors ESE-1 (transcription factor ETS 3), ELF3 (E74 like ETS transcription factor 3), Grhl2/Grhl3, Ovol1/Ovol2, GATA3/GATA4, FOXA1/FOXA2 (forkhead box), and KLF4 (kruppel−like factor 4) promote the epithelial phenotype in cells (Figure 1). Interestingly, these transcription factors are capable of repressing EMT or even inducing the reverse process called mesenchymal-epithelial transition (MET) in various cell models, promoting the epithelial phenotype [48,49,50,51,52]. 

Other biomolecules responsible for promoting the epithelial phenotype are epigenetic factors such as microRNAs (miRNAs), noncoding RNAs of approximately 22 bp, which negatively regulate gene expression at the post-transcriptional level by binding to mRNA and inhibiting its translation into proteins. One of the most relevant and most frequently studied is the miR-200 family (miR-200a, -200b, -200c, -141, and -429) which were initially described as negative regulators of ZEB1/ZEB2 expression [49,53,54]. Currently, miR-1, miR-145, miR-204, and mir-205 participation in the maintenance of epithelial status has been confirmed [50,55,56,57,58]. 

Long non-coding RNAs (lncRNAs) are RNAs of >200 bp with no apparent protein coding potential and have recently been described as epithelial and mesenchymal markers during EMT [59]. Several studies have shown that lncRNAs modulate the different EMT states at different levels, from the transcription and localization to the stability of proteins, and can be classified as suppressors or promoters of EMT [60]. LncRNAs act as competitor endogenous RNAs (ceRNAs) that sequester and inhibit miRNAs that promote or inhibit EMT, the most-studied epithelial lncRNAs (EMT suppressors) are GAS5 (growth arrest-specific 5), DREH (down-regulated in hepatocellular carcinoma), TUSC7 (tumor suppressor candidate 7), NBR2 (neighbor of BRCA1 LncRNA 2), and NEF [59,60]. In the EMT process, mesenchymal lncRNAs have been studied more.

### 2.2. Mesenchymal Markers

A reorganization of the cytoskeleton, the formation of focal adhesions and of invadopodia, a gain in the expression of intermediate filaments such as vimentin and cytokeratin 14, and a gain in the expression of α-SMA (alpha−smooth muscle actin) are characteristic of the mesenchymal phenotype and are; therefore, considered mesenchymal markers during EMT [61,62,63,64]. Furthermore, the formation of actin stress fibers that provide the posterior–frontal polarity of the cells is also considered a marker of the mesenchymal phenotype [42]. Cell migration through the extracellular matrix requires changes in membrane molecules, necessary for the formation of focal adhesions, multimolecular structures that bind the cytoskeleton with extracellular matrix proteins [65]. One of the most relevant components of focal adhesions is the integrins, heterodimeric transmembrane receptors composed of α and β subunits; and the gain of αvβ6, αvβ3, and α5β1 heterodimers are considered the most relevant mesenchymal integrins [61,66,67,68].

On the other hand, the remodeling of components of the extracellular matrix is essential for locally and distantly invasive processes whereby matrix metalloproteases (MMPs), zinc-dependent endopeptidases, are responsible for degrading components of the extracellular matrix. The increase in the expression, secretion, and activation of MMP-1, MMP-2, MMP-9, and MMP-10 is considered a mesenchymal marker of EMT [62,69,70]. in particular, MMP-14 (Membrane type 1-MMP) is located on the cell surface and forms part of the invasive front of invadopodia, which are membrane structures essential in the local invasion of mesenchymal cells; therefore, the increase in the expression, localization and activation of MMP-14 is a mesenchymal marker of EMT [69,70].

One of the most relevant membrane proteins is N-cadherin, which is part of the so-called “cadherin switch” in EMT, and is a protein related to migratory and invasive capacity [71]. On the other hand, the transmembrane glycoproteins of the mucin family, such as mucin 1, 4, and 16, are mesenchymal markers and their expression promotes an increase in migratory and invasive capacity [56,72,73]. Interestingly, the expression of CD44, another transmembrane glycoprotein, is also related to the stem and mesenchymal phenotype [47,64].

Various studies in cell models show that the transcription factors Twist1/2, Snail, Slug, and ZEB1/2 are master regulators of EMT, since the overexpression of some of these transcription factors is capable of inducing EMT, and are considered canonical markers of EMT [74,75]. Interestingly, these factors play a dual role at the transcriptional level, repressing epithelial markers and transcribing mesenchymal markers, promoting the initiation and acquisition of EMT [76]. Another mesenchymal transcriptional factor is β-catenin, a protein that forms a stable cell–cell adhesion complex in epithelial cells together with E-cadherin [75]. However, in mesenchymal cells the loss of E-cadherin allows for the release of β-catenin from this complex and subsequent translocation to the nucleus, allowing for the formation of a transcriptional complex with TCF/LEF and thus regulating the expression of genes related to the acquisition of the mesenchymal phenotype [77,78]. Of these factors, TCF3 and LEF1 have been the most studied and are established as markers of the mesenchymal phenotype [42,74,78]. 

Interestingly, it has been reported that a consequence of EMT is the acquisition of stemness characteristics such as an increase in the population of stem cells with a CD44^high^/CD24^low^ phenotype [75]. In this context, various transcription factors related to the stem phenotype such as SOX4/SOX9, Oct4, and Nanog have been found to be mesenchymal markers in various cell models [42,47,56,78]. However, various transcription factors are recognized as mesenchymal markers active in different stages of EMT such as STAT3 [79], E2F1 [80], AP-1 [42,47], and Smad [42,62].

Within epigenetic regulation, various miRNAs have been classified as mesenchymal markers; among these, miR-10b, miR-21, miR-27a, miR-29a and miR-221/222 can promote EMT when they are overexpressed. However, the most relevant are miR-21 and miR-221/222 since they regulate a large number of EMT-related targets [44,57,81,82,83,84,85]. In contrast to epithelial lncRNAs, mesenchymal lncRNAs have been described more often: NEAT1, MALAT1 (NEAT2), HOTAIR, MEG3, H19, ROR, and ATB have been identified as master regulators of EMT-related transcription factors such as Snail, Slug, ZEB, and Twist [59,60,86]. However, the mechanisms and the number of lncRNAs that promote EMT continue to be identified, so the biological relevance they play in EMT must continue to be studied.

## 3. Extracellular Signals Driving EMT 

Different extracellular signals are known to drive EMT and promote tumor progression. These signals can be classified as: Growth factors such as TGF-β and EGF [7,8,9,10,11]; cytokines such as IL-6, IL-8, and TNF-α [17,18,87,88,89]; chemicals as cadmium, sodium arsenite, Nickel(II) chloride, and Nickel(II) sulfate [19,20,90]; fatty acids such as palmitate and oleic acid [21,22,91]; hormones such as L-thyroxine (T4), 3,5,3′-triiodo-L-thyronine (T3), progesterone, and estradiol [23,24,25,92]; and others such as LOX-1 and ROS (reactive oxygen species) [26,27,93].

In different types of in vitro cancer models, such as colorectal, pancreatic, lung adenocarcinoma, gastric, hepatocellular, and others, these inductors promote the expression and activation of transcription factors (TF) such as Twist, ZEB1/2, Snail, Slug, and HIF-1α. These TF promote the expression of mesenchymal markers such as vimentin, N-cadherin, α-SMA, fibronectin, FOXC2, and cytokeratin5/8, and decrease epithelial markers such as E-cadherin and claudin-1; in addition, they increase the MMP-2 and MMP-9 secretion (Table 1). Together, these events promote the loss of cell–cell junctions and induce the spindle-shaped morphology [7,8,9,10,11,17,18,90]. Various signaling pathways regulate these biological processes during tumor progression, such as the Wnt-β-catenin, Notch-1, ALK5/H2O2, IL-8-JAK2-STAT3, Src-ANXA2-STAT3, and IKKα–Smad signaling pathways [7,9,10,19,20,87]. However, in recent years the role of leptin as an inducer in EMT, as well as in tumor progression, has become very important.

## 4. Leptin

Leptin is a protein with a molecular weight of 16 kDa, 167 aa and encoded by the *OB* gene (7q31.3) in humans [95]. Leptin is a hormone secreted mainly by adipose tissue; through the endocrine system, it maintains the homeostasis between energy consumption and expenditure. Serum leptin levels are directly proportional to the amount of adipose tissue and both increase progressively with obesity [96,97,98]. Obesity is a serious public health problem worldwide. Etiological and pathophysiological studies of obesity establish that leptin is a central biomolecule for the development of this pathology, although the molecular mechanisms mediated by this adipokine have not yet been fully described [99,100].

Since its discovery in 1994, leptin has first been linked to important physiological processes primarily related to satiety, reproduction, and metabolic adaptation as an evolutionary response to starvation [30,98,101,102]. Over the years, research on the biological function of leptin has increased, showing great interest in the discovery of a possible treatment for obesity and other pathologies mainly related to the reduction of this hormone through combinations of biomolecules or the replacement of leptin [100,103,104].

In recent years it has been reported that leptin also plays a significant role within the tumor microenvironment, especially when its expression is chronically elevated, contributing to the development and progression of different types of cancer. The population at potential risk are patients with overweight or obesity [35,105,106]. In this context, leptin acts as a pro-inflammatory cytokine in the tumor microenvironment, exerting endocrine, paracrine, and autocrine action, inducing the activation of cancer-associated fibroblasts (CAF) and, high levels of MMPs, and promoting cell infiltration to the immune system such as via cancer-associated macrophages (CAM), triggering the secretion of pro-inflammatory and pro-angiogenic cytokines and growth factor [107,108,109]. In tumor cells and within the tumor microenvironment, the over-expression of leptin and its receptor, ObR, promotes proliferation and the EMT activation program, a biological process that promotes invasion and metastasis [38,108,110,111,112,113].

Less than a decade has passed since leptin was proposed as a new inducer of EMT [77]. Since then, many studies have examined the relationship between leptin and EMT in various types of cancer; they will be described below.

### 4.1. Signaling Pathways

Currently, several leptin signaling pathways have been reported to be activated by the interaction with its membrane receptor, Ob−Rb [114]. Leptin belongs to the class I long-chain superfamily of cytokines; it has a compact tertiary structure, characterized by two pairs of antiparallel alpha helices, and a disulfide bridge that involves two cysteine residues (Cys 96 and Cys 146), fundamental for binding with its receptor [97]. The Ob−R is encoded by the *LEPR* gene (1p31.3), which belongs to the family of cytokine receptors for glycoprotein 130 (gp130) and the six isoforms ObR-a, ObR-b, ObR-c, ObR-d, ObR-e, and ObR-f. These variants are expressed in most tissues and share 78% homology between mice and humans [41,115,116,117,118]. All six isoforms share an extracellular domain of 862 aa, a N-terminal domain composed of two homologous cytokine receptor domains (CRH1 and CRH2) separated by an immunoglobulin (Ig) domain and two fibronectin type 3 (FNIII) domains close to the transmembrane region [115]. The study of the conformational dynamics of the extracellular region of the ObR, bound and not bound to cytokines, shows that the CRH2 domain is essential for binding with the leptin epitope (II) while the Ig domain stabilizes the ligand by interacting with its epitope (III) and with the FNIII domains in the initial activation of ObR-b [119,120]. With the exception of the secreted ObR-e isoform that functions as a free circulating leptin transporter, modulating its half-life and availability to tissues, the rest of the isoforms present a 34 aa transmembrane domain and a proline-rich motif, called box-1, in its intracellular region [105]. The long ObR-b isoform is the only isoform that contains in its intracellular sequence a second hydrophobic interaction motif described as box 2, and several phosphorylation sites that recruit and bind to Janus-activated kinase (JAK), responsible for fully activating signaling pathways [121]. 

In the absence of leptin, ObRs are found as complexes of inactive homodimers and homo-oligomers [122]. The binding of leptin to its receptor causes a conformational change in the extracellular domain, forming quaternary complex dimers with 2:2 stoichiometry, although a second superior oligomerization model has also been proposed that involves 4:4 stoichiometry—that is, formed by two dimers of the Lep–ObR complex—although these data are still under discussion (Figure 2) [119].

### 4.2. Canonical Leptin Pathways

As mentioned previously, leptin induces the activation of signaling pathways through interaction with its receptor. The first response is through the JAK2 kinase; so far, three canonical signaling pathways related to leptin functions have been established [123].

The first canonical pathway activated by leptin is the JAK-STAT pathway, related to the regulation of food intake and body weight [124]. The STAT family in mammals includes seven members, comprising STAT1–4, STAT5a, STAT5b, and STAT6, with STAT3 and STAT5 as the main mediators of the leptin-induced anorexigenic effects [123,125]. In addition, protein 1, related to the low-density lipoprotein receptor-related protein 1 (LRP1), binds directly to the receptor–leptin complex and contributes to the phosphorylation of ObR-b and the activation of STAT3 [126]. 

The onset of the activation of the JAK2-STAT3 signaling pathway occurs with the phosphorylation at Tyr 1138, the recruitment of STAT3 through its SH2 domain (Src homology 2), its phosphorylation at Y705 and its subsequent dimerization and translocation to the nucleus, where it induces gene transcription of the appetite regulating-neuropeptides POMC, AgRP, and NPY and the suppressor of cytokine signaling 3 (SOCS3) [114]. Additional factors involved in the negative regulation of this pathway are tyrosine phosphatases such as PTP1B, which inhibits JAK2-STAT3 signaling by directly dephosphorylating JAK2, while the tyrosine phosphatase TCPTP blocks phosphorylation at Y705 in STAT3, necessary for dimerization and translocation to the nucleus [123,127]. On the other hand, the JAK2-STAT5 pathway initiates the phosphorylation of JAK2 at Tyr1077, followed by activation and translocation to the nucleus of STAT5 [128]. Deletion of STAT5 has been shown to cause leptin resistance, hyperphagia, and obesity, contributing to leptin-dependent regulation of energy balance and body weight [129].

The mitogen-activated protein kinase (MAPK)/extracellular signal-regulated protein kinase 1/2 (ERK1/2) pathway participates in the transcription of genes involved with cell proliferation, differentiation, survival, and apoptosis [130]. The ERK pathway activation depends on the JAK phosphorylation at Tyr985 promoting the interaction between Src homology region 2 domain-containing phosphatase-2 (SHP2), and growth factor receptor-bound protein 2 (GRB-2), resulting in ERK activation and inducing its translocation to the nucleus to mediate the gene expression of c-Fos and of the early growth response protein, EGR-1 [129,130].

Another canonical leptin pathway is IRS-PI3K, whose activation is regulated by the recruitment of IRS1/2 to JAK2 through the SH2B1 domain, promoting its phosphorylation and subsequent recruitment of the regulatory subunit p85 of PI3K, and promoting phosphorylation of phosphatidylinositol-4, 5 -biphosphate (PIP2) to phosphatidylinositol-3, 4, 5-triphosphate (PIP3) in its PH domain, consequently triggering the activation of protein kinase B (PKB), also known as Akt [123,131,132]. On the other hand, the phosphatase and tensin homolog (PTEN) inhibits PI3K by dephosphorylating PIP3, inhibiting the activation of downstream molecules such as Akt [128].

Leptin signaling also induces a negative feedback loop mediated by SOCS3 through the binding of its SH2 domain to Tyr985 of JAK, inhibiting the MAPK pathway and dephosphorylating JAK2 in a negative feedback loop of the leptin–ObR signal as well as proteins PTP1B and TCPTP, molecules directly associated with leptin resistance [114,133,134,135,136]. SHP2 has also been reported to lead to decreased phosphorylation of JAK2, and; therefore, regulates the termination of leptin-initiated signal transduction [137].

### 4.3. Non-Canonical Leptin Signaling Pathways 

The IRS/PI3K signaling pathway activates cyclic nucleotide phosphodiesterase 3B (PDE3B) enzymes, whereby PDE3B promotes the decrease in levels of cyclic adenosine monophosphate (cAMP) which inhibits the activity of the cAMP response element binding protein (CREB) by suppressing NPY expression through the cAMP–CREB pathway to achieve the anorexigenic effect of leptin. Furthermore, the PI3K–PDE3B–cAMP pathway also interacts with the JAK2–STAT3 pathway, reducing food intake and preventing overweight and obesity [138]. On the other hand, the forkhead box protein O1 (FoxO1) is phosphorylated by Akt kinase; once activated, FoxO1 translocates to the nucleus, acting as a transcription factor in the expression of the orexigenic genes NPY/AgRP and suppressing the expression of POMC, the constitutive expression of the Akt–FoxO1 pathway, which results in a loss of the ability of leptin to inhibit food intake [139].

Furthermore, the activation of Akt triggers the activation of the mTOR/S6K/AMPK signaling pathway, where mTOR activates S6K, an inhibitor of AMPK [140,141]. AMPK is formed by a catalytic α subunit and two regulatory subunits (β and γ); its activation depends on the phosphorylation at Thr172, while its inhibition is mediated by the phosphorylation at Ser485/49 [140]. AMPK phosphorylates and inactivates acetyl-CoA carboxylase (ACC), a key enzyme in fatty acid biosynthesis. Inhibition of hypothalamic AMPK is sufficient to reduce food intake and gain weight [142]. Heat shock protein (Hsp60) has been reported to activate ERK1/2, c-Jun NH2-terminal kinase (JNK), MAPK p38, and nuclear factor (NF)–κB, as well as Akt phosphorylation, stimulating the secretion of TNF-α, IL-6, and IL-8, which are related to inflammatory processes in adipose tissue [143].

In the absence of JAK2, leptin activates the STAT3 and MAPK pathways through the Src tyrosine kinase family [144]. Recently, we reported that leptin promotes the expression of proteins involved in EMT through the FAK-Src; pathway; therefore, there is a relationship between the activity induced by leptin and Src [40,145,146,147].

In humans, plasma leptin concentrations range from 1.5 ng/mL in healthy patients to 8.6 ng/mL in obese patients, showing that in obese people there is an increase in serum leptin concentration [148]. In vitro studies indicate that the response of leptin at low doses is physiologically normal and the response triggered at higher doses of leptin results in the alteration of signaling pathways and their association with pathological processes [149]. In addition to elevated leptin levels, some studies indicate that mutations in leptin, ObR, and signaling pathways are important objects of study to comprehend multiple diseases such as cancer [150].

Future research should focus on the conformational changes of intracellular signaling altered by elevated levels of leptin and their relationship with pathological mechanisms in cancer.

## 5. Role of Leptin Driving EMT in Breast cancer 

Breast cancer is the most common cancer in women, with the highest rate of incidence and mortality worldwide [151]. In addition to genetic, hormonal, reproductive, and lifestyle risk factors, obesity is an important area of research due to its key role in the tumor microenvironment and in the induction of EMT, which is an important feature of tumor progression in breast cancer. Several studies have evaluated the effect of leptin on the expression or activation of biomolecules related to EMT and the acquisition of mesenchymal characteristics. These investigations were conducted using in vitro and in vivo models, and there are also reports using patient samples (Figure 3A). The findings are described below.

### 5.1. In Vitro Assays

The direct effect of leptin on the induction of EMT has been demonstrated using different normal and cancerous epithelial cell lines. We evaluated the effect of leptin as a promoter of EMT in the MCF10A non-tumor mammary epithelial cell line. In 2017, we found that, in MCF10A cells, leptin promotes EMT markers, such as morphological changes from a cuboid to a fusiform phenotype, vimentin expression, and relocation of E-cadherin to the cytoplasm. In this cell model, ERK and FAK kinases were found to regulate EMT-related changes and leptin-induced cell migration in MCF10A cells [146]. Further investigations showed that leptin, through the activation of Src and FAK kinases, regulates the expression and subcellular localization of the EMT-related transcription factors Twist and β-catenin; it also induces a significant secretion of MMP-2 and MMP-9, and promotes invasiveness, all of which are characteristics related to the activation of the EMT program [152]. Src and FAK kinases play an important role in the regulation of EMT in the MCF-7 and MDA-MB-231 breast cancer cell lines, where leptin promotes MMP-2 and MMP-9 secretion, cell migration, and invasion through the activity of Src and FAK, and PI3K-activity independent [147]. In the noninvasive breast cancer epithelial cell lines MCF-7 and ZR-75-1, it was demonstrated that leptin, through the JAK/STAT3–Akt signaling pathway, can downregulate CCN5 (Cyr61, CTGF, -Nov), promoting EMT activation through the reduction of E-cadherin, the overexpression of vimentin and Snail, and the stimulation of cell migration [153]. In MCF-7 and SK-BR-3 breast cancer cells, Wei et al. found that leptin, through a mechanism dependent on the activation of the canonical PI3K/Akt pathway and the overexpression and activation of pyruvate kinase M2 (PKM2), promotes the expression of vimentin, fibronectin, and Twist and the downregulation of E-cadherin [111]. Using the same in vitro models, MCF-7 and SK-BR-3, it was demonstrated that the leptin-PI3K-Akt axis also regulates the expression of IL-8, a cytokine that contributes to the leptin-induced high levels of vimentin and fibronectin and the increase in cell migration and invasion [154]. Furthermore, in 2016, a co-culture demonstrated the collaborative role of M2 macrophages, derived the THP1 from human monocyte cell line, in the acquisition of mesenchymal characteristics in breast cancer cell lines. Leptin-MAPK-ERK1/2 and p38/MAPK lead to IL-8 promoter activation and IL-8 mRNA and protein expression in M2 macrophages. Consequently, IL-8 derived from leptin-treated macrophages induces cell migration and invasion in MCF7 and MDA-MB-231 cells [110].

In MCF-7 cells, leptin promotes metastatic potential and stemness by inducing morphological changes to a mesenchymal phenotype, with a greater rate of migration and invasion and a decreased epithelial phenotype via TGFβ1 [112]. In MCF-7 and MCF10AT1 breast cancer cells, leptin downregulates *CDH1* expression and upregulates *SNAI2* expression through an autocrine mechanism that involves the expression and secretion of TGFB1 [112]. In the MCF-7, SK-BR-3, MDA-MB-468, and MDA-MB-231 cell lines, leptin induces the expression of vimentin and, fibronectin and the downregulation of E-cadherin [111,154].

In two cell lines derived from MMTV-Wnt mice (E-Wnt and W-Wnt) and the human MDA-MB-231 breast cancer cell line treated with serum collected from mice with diet-induced obesity, leptin induced EMT features such as greater cell migration, invasion, and overexpression of *Akt3*, *Cdh2*, *Foxc2*, *Vim*, *Twist1*, and *Twist2* in all cell lines [155]. In HMT-3522 S1 non-neoplastic mammary epithelial cells, leptin regulates the relocalization of ZO-1 from cell–cell junctions, and the loss of occludins and claudins from tight junctions in a PI3K-Akt-dependent pathway [156]. 

In MCF-7 and T47-D cells, it was also observed that Acyl-CoA: (cholesterol acyltransferase, *ACAT2)*, a membrane-anchored enzyme involved in the production of cholesteryl ester, contributes to the leptin-induced migration and invasion rate through the PI3K-Akt pathway [157]. This study demonstrates the crosstalk between leptin induced pathways and fatty acid metabolism. 

In an in vitro model of differentiated acini derived from HMT-3522 S1 mammary epithelial cells, and in 3D cultures of nonimmortalized post-stasis human mammary epithelial cells (psHMEC), leptin leads to apical–basal polarity loss by modulating the distribution of the tight junctions (TJ) proteins ZO-1 and claudin-1 and unsettling the cortical F-actin cytoskeleton [156]. Additionally, in the acini models, leptin also affects the normal distribution of the adherens junction (AJ) proteins E-cadherin and β-catenin [156].

Zheng et al. demonstrated the determinant role of leptin signaling in maintaining the mesenchymal phenotype by silencing the leptin receptor in MDA-MB-231 cells. They observed a rounded epithelial-like shape, decreased vimentin expression, and 300-fold increased expression of E-cadherin [158].

In MCF-7 epithelial breast cancer cells, leptin promotes morphological changes to a fibroblast-like appearance along with invadopodia formation and stress fibers, as well as high levels of fibronectin, N-cadherin, vimentin, Snail, Slug, Zeb1, and Twist; Occludin and E-cadherin were also downregulated [77]. In MDA-MB-468 and MDA-MB-231 cells, which are more aggressive breast cancer cell lines, an increase in fibronectin, Slug, and Snail was observed, as well as a higher rate of cell migration and invasion; and a low expression of cytokeratin-18 [77]. These authors suggest that leptin partially regulates these events through MTA1/Wnt1 and Akt activation, which leads to GSK3β phosphorylation and consequent β-catenin cytoplasm accumulation and nuclear translocation, in combination with the association to TCF/LEF-binding sites from target genes such as cyclin D1 and fibronectin [77]. 

In nontumor MCF12A mammary epithelial cells and MCF7 breast cancer cells, leptin induces a mesenchymal phenotype, ZEB1 expression, and a “switch” from E-cadherin to N-cadherin, as well as the classic CD24^-^/CD44^+^ stem signature, and a larger number of spheres through STAT3 [159]. Furthermore, leptin activates STAT3 to act as a transcription repressor by interacting with G9, an H3K9 methyltransferase. The STAT–G9 complex represses the miR-200c epithelial maintenance-related noncoding RNA by binding to its promoter region. Thus the leptin–STAT3/G9–miR–200c axis regulates the increase in EMT and expression on CSC characteristics by direct enhancement of ObR expression [159].

### 5.2. In Vivo Assays

To further evaluate the role of leptin in the induction of the EMT program in vivo, several study models have been used. In MMTV-Wnt-1 transgenic mice with diet-induced obesity (DIO), it was found by using RNA-sequencing (RNA-seq) and Ingenuity Pathway Analysis (IPA) that leptin acts as a master regulator of the EMT gene signature by up-regulation of *AKT3*, *PIK3R1*, *TWIST1*, *TWIST2*, *WNT11*, and *WNT9B.* These mice also showed an up-regulation of the *AKT3*, *WNT11*, *FRAT1*, and *WNT9B* genes from the Wnt/β-catenin signaling, which is also related to the EMT program. Consequently, DIO mice showed major expression of *Akt3, Pik3rl, Twist1* and *2, Foxc2,* and *Vim;* as well as downregulated *Cdh1*. Gene expression was validated by immunodetection of high levels of vimentin [155].

Xenograft assays were performed in leptin-treated nude mice, where leptin induced more and weightier tumors, decreased the survival time and promoted lung and liver metastasis. Levels of Ki67, IL-8, and vimentin were higher and E-cadherin was diminished [154]. Furthermore, Cao et al. demonstrated the pivotal role of macrophages in the leptin-induced effects in nude BALB/C mice, where leptin promotes the largest tumors and pulmonary metastasis, reduces survival, and also induces IL-8 overexpression [110].

In accordance with the in vitro determinations, Yan and collaborators showed that in breast tumors from athymic nude mice, leptin induces Wnt1 expression and modulates GSK3β phosphorylation and β-catenin nuclear accumulation as well as vimentin expression [160]. Using obese Sprague–Dawley rats, they found a high incidence of more aggressive tumors, with overexpression of ObR, leptin, and p-STAT, and a reduced expression of miR-200c. Similarly, STAT3 inhibition diminished the mammary carcinoma incidence and decreased the tumor size and aggressiveness; also, minor ObR expression and high levels of *miR-200c* and *Cdh1* were observed in tumor tissue [159].

### 5.3. Patient Samples 

In addition to several reports indicating that high concentrations of circulating or local interstitial leptin are related to poor prognosis for breast cancer patients, some of the molecules involved with the leptin-induced pathways driving EMT, as previously studied in in vitro and in vivo models, were found in high levels in breast cancer patients tissue or serum samples.

In breast cancer patients with a high body mass index (BMI) and elevated circulating levels of leptin, the conditioned media from adipose explants showed a high leptin concentration of 57.4 ± 24.1 ng/mL, there was a tissue concentration of 22.18 ± 15.24 ng/mL for overweight patients, and 22.8 ± 10.63 ng/mL for obese patients [156]. Testing 164 breast cancer patients’ samples, Laforest et al. found that mammary adipocyte size is positively correlated with tumor, metastasis (TMN) stage, and tumor grade, independently of ER, PR, and HER2 expression status. In addition, they showed a positive correlation between breast fat cell diameter and *IL6*, *TNF,* and *LEP* expression [161]. In a different study, of 114 breast carcinoma samples, a slight overexpression of leptin was observed in 79.8% and ObR-b in 85.1%; a strong positive expression was seen in 21.1% (leptin) and 11.4% (ObR-b) of the samples, independently of age, menopausal condition, pathological type, and ER or PR status. Furthermore, leptin expression was strongly correlated with poor overall and disease free survival rates [162]; a similar but sightly trend has been described for ObR-b expression [38,162]. 

Once we have established that obesity is related to breast cancer malignancy, it is important to show how this relationship occurs. So, to establish leptin’s role in normal tissue is a good approach; in normal breast tissue samples with high leptin levels, there is an atypical distribution of epithelial markers which suggests a loss of apico-basolateral polarity [156]. Consequently, high levels of ObR, leptin, IL-8, and vimentin, as well as decreased E-cadherin, were detected in tissue samples of in situ and invasive breast cancer when compared with benign hyperplasia [154]. Additionally, invasive breast carcinoma tissues expressed high levels of ObR, CD68, and IL-8; these molecules participate in the paracrine mechanisms from the tumoral microenvironment occurring between macrophages and tumoral cells [110].

In accordance with the in vitro and in vivo analysis carried out by Chang and collaborators, a microarray analysis using normal and malignant breast tissue samples, found a strong correlation between high ObR, leptin, and p-STAT expression and low miR-200c expression, with poorly differentiated high-grade tumors. Furthermore, in 11 patients’ samples, the enrichment of STAT3/G9a was established at the miR-200c promoter in TNBC, which involves are more aggressive tumors in comparison with luminal breast cancer [159].

## 6. Role of Leptin Driving EMT in Gastric Cancer

### 6.1. In Vitro Assays

Leptin promoted STAT3 and ERK 1/2 activation in the AGS, SNU-484, SNU-601, and SNU-638 gastric cancer cell lines [163]. Interestingly, in MKN28 and HS746T gastric adenocarcinoma cells, leptin induces STAT3 and ERK2 activation in a JAK-dependent pathway; furthermore, ERK2 activation is also required for SHP2 phosphorylation and Grb2 association. Both of these signaling axes were important for the leptin-induced proliferation of gastric cancer cells (Figure 3E) [164]. In the MKN28 gastric cancer cell line, leptin treatment induced JAK2 and ERK1/2 activation through transactivation of the EGFR, which in turn seems to be dependent on the metalloproteinase-dependent release of EGFR ligands [165].

### 6.2. In Vivo Assays 

Mice whose gastrointestinal epithelial cells displayed SOCS3 deficiency (T3b-SOCS3 cKO mice) were treated with leptin and a sustained STAT3 phosphorylation was observed in gastric mucosa. It was also found that, from three to eight weeks of age, there is a progressively stronger expression of leptin, as well as greater ObR and STAT phosphorylation, compared with the mucosa from control mice [166]. After eight weeks of leptin treatment, cKO mice presented enlarged stomachs with multiple proliferating adenocarcinomas, and after 10 weeks infiltrated cells increased in interglandular and submucosal spaces. Furthermore, in the carcinomatous epithelium of cKO mice, β-catenin nuclear accumulation, decreased E-cadherin, and an increase in Snail expression were observed. Leptin signaling blockade with a specific antibody reduced the STAT3 hyperactivation and the hyperproliferative phenotype [166]. In addition, STAT3 hyperactivation is a key step for tumor development in the gp130 757F/F mouse model of gastric tumorigenesis, as demonstrated by a variant of mice displaying reduced STAT3 activity (gp130 757F/Y757F: STAT3+/−) showing negligible tumor development, inflammatory infiltration, angiogenesis, and expression of MMPs [167].

In male C57BL/6J mice fed a high-fat diet (HFD) for three weeks, high leptin and ObR expression in gastric mucosa was observed, as well as perinuclear β-catenin accumulation, furthermore, β-catenin accumulation did not occur in leptin-deficient ob/ob and ObR-mutated db/db mice. Presumably, β-catenin expression occurs downstream of ObR-PI3K-Akt signaling in this HFD mouse model [168]. A similar effect occurred with the high leptin expression and ObR, STAT3, Akt, and ERK phosphorylation observed in HFD WT mice; responses diminished in ob/ob and db/db mice [169]. In 2019, Arita et al. further explored the implications of HFD-induced leptin expression and ObR phosphorylation, showing that both are correlated with intestinal metaplasia pathogenesis [170]. 

### 6.3. Patients

In patient samples, Pai et al. also observed high leptin and its ObR receptor expression in the basement membrane of gastric cancer samples when compared with normal gastric tissue, where leptin and its receptor were only observed in cells of the progenitor zone [164]. The expression of ObR and leptin was also evaluated by immunostaining in 207 gastric carcinoma samples. They found positive ObR expression in 67 carcinoma samples, of which 45 showed strong leptin expression and 22 weak leptin expression; both ObR-b and leptin expression were correlated with the depth of tumor invasion, venous and lymphatic metastasis, and TMN stage. Furthermore, metastasis was observed in 10.9% of patients with high levels of leptin [171]. In patients with intestinal metaplasia (IM), the serum leptin concentration media was 116.6 pg/mL and, by multivariate analysis, it was found that leptin concentration is a risk factor for diagnosis of IM, which in turn serves to identify patients at risk of developing gastric cancer [172]. In 38 samples from patients with gastric adenoma (GA), early gastric cancer (EGC), and advanced gastric cancer (AGC) assessed by immunostaining for leptin and ObR expression, leptin was expressed in >40%, while ObR expression showed a stage-related progressive increase from 2.6% to 18.4% [163].

## 7. Role of Leptin Driving EMT in Prostate Cancer 

Serum leptin levels have been reported to be higher in prostate cancer patients than in healthy patients [173]. In addition, leptin expression in tissue is significantly higher in patients with localized and metastatic prostate cancer compared to benign hyperplasia [174]. In 2011, an excess of adipose tissue and high serum leptin levels were directly corelated to prostate cancer aggressiveness, rather than being seen as risk factors [175]. Later, obesity-related systemic factors that promote an invasive phenotype, including an increase in proliferation, invasion, cell migration, EMT, and matrix metalloproteinase activity in prostate cancer cells, were demonstrated [176].

Leptin promotes EMT by inducing morphological changes from an epithelial to a mesenchymal phenotype, decreasing the expression in E-cadherin and increasing the vimentin expression in benign prostatic hyperplasia epithelial-1 (BPH-1) cells through the downregulation of bone morphogenic protein and activin membrane-bound inhibitor (BAMBI) [54]. Leptin promotes EMT and cell migration by stimulating the STAT3 pathway in DU-145 and PC3 prostate cancer cells [177]. One of the functional consequences of EMT is chemotherapy resistance and the leptin-Notch axis impairs the 5-fluorouracil effect in BxPC-3 and MiaPaCa-2 pancreatic cancer cells [178]. Additionally, in DU145 cells, leptin promotes proliferation, invasion, and inhibition of apoptosis through activation of the ERK1/2 pathway [179]. On the other hand, in PC3 cells, leptin induces an increase in the migration and expression of the mesenchymal integrin αvβ3 through ObR-IRS-1-PI3K-Akt-NF-κB signaling [180]. In LNCaP, DU145, and PC-3 cells, it was observed that chronic treatment with leptin increases cell proliferation, migration, and invasion by inactivating FoxO1 through PI3K-Akt signaling, in addition to increasing the expression of the leptin receptor (Figure 3B) [181].

## 8. Role of Leptin Driving EMT in Lung Cancer

Despite the few studies linking leptin to the progression of lung cancer, it has been reported that the polymorphism of the leptin gene LEP-2548 G/A increases the susceptibility to the development of non-small-cell lung cancer (NSCLC), which represents 75–80% of lung cancer [172]. Additionally, it is suggested that the elevated serum leptin concentration in patients with stage I to III NSCLC plays an important role in the progression of this type of cancer [182]. In addition, leptin and its ObR receptor have been related to the development and progression of NSCLC, since the overexpression of leptin and its receptor is significantly higher in NSCLC tissues compared to normal lung tissues, and leptin is considered an indicator of poor prognosis in the development and progression of NSCLC [126]. These data suggest that the levels of leptin in both serum and lung cancer tissue have a clinical prognostic value for the development and progression of lung cancer; however, it is necessary to continue pursuing studies that will help us to understand the role that exerts leptin in the tumor progression of NSCLC [4].

Leptin promotes invasion and metastasis by inducing EMT in A549 lung cancer cells by the upregulation of mesenchymal markers such as vimentin, N-cadherin, and Twist, and the downregulation of epithelial markers E-cadherin and β-catenin [183]. Leptin also induces EMT and expression of the transcription factors ZEB and Twist via activation of the ERK signaling pathway in A549 lung cancer cells [184]. Interestingly, in the tumor microenvironment, leptin and its ObR receptor are overexpressed and leptin produced by cancer-associated fibroblasts promotes proliferation and cell migration in non-small-cell lung cancer (NSCLC) cells, conceivably via the PI3K-Akt and MAPK/ERK1/2 signaling pathways in a paracrine manner [185]. 

The elevated leptin level in the patient’s serum is correlated with the severity of lung fibrosis and leptin and significantly promotes EMT in A549 lung cancer cells, as evidenced by increased collagen I and α-SMA expression. Leptin also accelerates EMT through inhibiting autophagy via the PI3K-Akt-mTOR pathway [177]. Interestingly, in humans the expression and clinical importance of leptin in lung cancer is a relevant since, in both serum and tissue samples, the leptin levels of the group of patients with lung cancer were significantly higher in comparison with those of the control group. Leptin was strongly correlated with sex, but not with other tumor-related factors (Figure 3C) [186]. 

## 9. Other Cancers 

Currently, there is little information about the leptin-induced molecular mechanisms that regulate the growth of cholangiocarcinoma cells; however, evidence from in vitro studies in HuH-28 intrahepatic cholangiocarcinoma cells indicates that leptin promotes cell proliferation and migration, while in in vivo models of cholangiocarcinoma cells, leptin promotes the activation of the STAT3 and ERK1/2 signaling pathways [187]. In human cholangiocarcinoma cell lines SK-ChA-1 and TFK-1, leptin stimulates EMT by stimulating cell migration and invasion, decreasing in E-cadherin and β-catenin expression, and increasing vimentin and, N-cadherin expression as well as the proangiogenic capability through the miR-122/PKM2 axis (Figure 3D) [188]. 

Leptin exposure increase cancer cell migration and invasion through the activation of the JAK-STAT3, PI3K-Akt, and RhoA-ROCK pathways, and promotes new lamellipodial, stress-fibers, and focal adhesion formation and the maintenance of stemness and the mesenchymal phenotype in ovarian cancer cell lines. In humans, serum and ascites leptin levels were higher in overweight patients, who had worse survival, and ObR-b was more highly expressed in ascites and metastasis than in primary tumors (Figure 3F) [37].

In the BG-1, OVCAR-3, and SKOV-3 ovarian cancer cells, the expression of the short and long isoforms of ObR was seen; furthermore, in BG-1 cells, leptin-dependent MAPK/ERK1/2 activation and proliferation were observed [189]. In OVCAR-3, the effect of leptin treatment in the activation of MEK/ERK1/2 and PI3K-Akt signaling pathways and posterior cyclin D1 and Mcl-1 expression was demonstrated [190]. In SKOV3 and OVCAR3 cell lines, leptin also promoted ERK1/2 and JNK1/2 activation, MMP-7 expression, and MMP-2/MMP-9 expression and activation [191]. A side effect was the activation of the JAK2-STAT pathway that was observed in HEY3 and SKOV3 ovarian cancer cells [192]. A collaborative effect of Erα and ObR activation resulted in MMP-9 overexpression, high proliferation, and invasion [36].

In samples from ovarian and endometrial carcinomas, ObR expression was detected, with higher levels in ovarian cancer [193]. Additionally, a poor prognosis for Paclitaxel/Docetaxel-treated patients was associated with leptin expression due to reduced chemosensitivity [194].

Overexpression of the leptin receptor ObR in patient tissue samples is associated with bladder carcinogenesis [195]. In UMUC3 and 647V human urothelial carcinoma cell lines, leptin promotes migration and expression of phospho-NF-κB, NF-κB, Snail, Slug, Y-box-binding protein 1, and COX-2 as well as NF-κB-vimentin-STAT3 signaling (Figure 3G) [196].

In Barrett’s esophageal adenocarcinoma cell lines BIC-1 and SEG-1, leptin has been shown to promote cell proliferation [197]. On the other hand, in OE33 cells of esophageal carcinoma, leptin promotes the activation of the JAK2, ERK, and Akt signaling pathways, which induce the expression and activation of cyclooxygenase-2 (COX-2), prostaglandin E2 (PGE2), EP-4 receptor, and EGFR transactivation, events that contribute to tumor progression [198]. Furthermore, in esophageal adenocarcinoma cells OE33, OE19, BIC-1, and FLO, it was shown that leptin induces the transactivation of EGFR through heparin-bound epidermal growth factor (HB-EGF) and transforming growth factor alpha (TGFα) through the activity of MMPs, events that promote cell proliferation [199].

In patients with esophageal adenocarcinoma, an increase in adipocyte size was directly associated with leptin expression, angiogenesis, and lymphangiogenesis. With nodal metastasis, interestingly, leptin increased the mRNA levels of two key regulator genes of EMT, α-SMA and E-cadherin, in OE33 esophageal adenocarcinoma cells [200].

The relationship between leptin and kidney cancer has been little explored. In a recent study, Perumal et al. did not find a significant difference in the expression of leptin and its receptor between clear cell renal cell carcinoma (ccRCC) tissue and healthy kidneys, nor between obese and nonobese individuals with cancer. Zhang et al. indicated that leptin expression is higher in healthy tissue, and Spyridopoulos et al. found that serum leptin levels were inversely associated with the risk of renal cell carcinoma (RCC) [44,201,202]. However, it has been reported that both serum leptin levels and the leptin receptor expression in tissue are higher in patients with renal cancer with venous invasion compared to patients without venous invasion, indicating a significant association with the histological type and metastasis to lymph nodes, as well as shorter disease-free survival in patients with serum leptin of ≥5.0 ng/mL [203]. In experimental models, in the Renca murine kidney cancer line, leptin increases invasiveness in a manner dependent on ERK1/2 kinase and Rho GTPase [203]. In Caki-2 primary kidney carcinoma cells, leptin increases proliferation and migration in a manner dependent on the JAK-STAT3 and ERK1/2 pathway [144]. Based on this, leptin-induced signaling could play a more significant role in renal cancer progression by promoting processes such as migration and invasion related to EMT.

On the other hand, the relationship between leptin and endometrial cancer has been little explored; however, it has been described that serum leptin levels in endometrial cancer are dependent on body mass index and that the decrease in leptin concentrations is significantly associated with a reduced risk of endometrial cancer [204,205]. In addition, the expression of leptin and its receptor is associated with lymph node metastases and a worse prognosis in endometrial cancer [14]. In the endometrial cancer cell lines An3Ca, SK-UT2, and Ishikawa, leptin regulates the expression of pro-angiogenic factors such as VEGF, IL-1β, LIF, and their respective receptors through the activation of JAK2-PI3K-ERK-mTOR, in addition to expressing higher levels of the leptin receptor (ObR-b and short isoforms) in endometrial cancer cells compared to benign primary endometrial cells [206]. In ECC1 and Ishikawa endometrial cancer cells, leptin promotes cell proliferation and invasion through the JAK2-STAT3, PI3K, ERK2, and COX2 pathways [135,207]. However, in 11Z, 12Z, and 22B endometriotic cells, leptin promotes cell migration and invasion in a manner dependent on the activity of the metalloprotease MMP2 and JAK2-STAT3 and ERK2 signaling [208].

Leptin promotes invasion and decrease of TIMP-1 and E-cadherin in a dose-dependent and time-specific manner in BeWo human choriocarcinoma trophoblast cells [209].

## 10. Conclusions

Leptin is an important protein in energy homeostasis, regulating food intake and energy expenditure; this process is also relevant during embryogenesis. However, high levels of leptin and its receptor are associated with tumor progression, so it has become an important actor in cancer. In recent years, leptin has been found to promote cell migration and invasion, the maintenance of cancer stem cells, the inhibition of apoptosis, and the induction of EMT in cancer cells. In in vitro models using several cancer cell lines, it was found that leptin promotes the expression of mesenchymal markers, a decrease in epithelial markers, and an increase in cell migration and invasion. On the other hand, using mice as in vivo model, leptin was shown to promote EMT through the WNT/β-catenin- dependent pathway, in addition to promoting a more aggressive phenotype of tumors, inducing metastasis to the lungs and reducing survival. Interestingly, in obese and overweight patients, elevated levels of leptin and its ObR-b receptor are associated with increased angiogenesis, metastasis, and a poor prognosis in cancer. All these characteristics demonstrate the significant role of leptin as a new and dynamic actor driving EMT and functional consequences in cancer. Based on these reports in different study models, it is suggested that leptin could be used as a potential tumor marker in the diagnosis and prognosis of lung, liver, colon, and especially breast cancer.

## Figures and Tables

**Figure 1 biomolecules-10-01676-f001:**
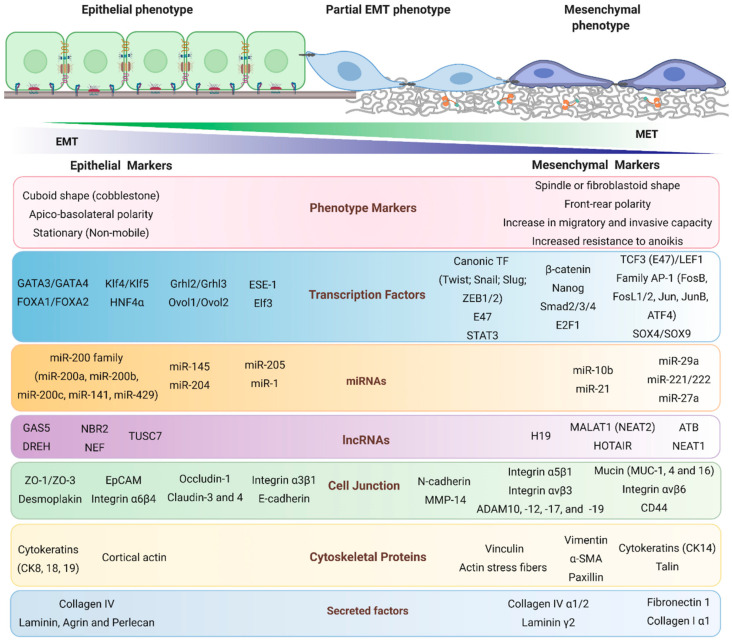
Biological markers of the epithelial–mesenchymal transition in cancer. The epithelial cells lose the cuboid nature and apico-basolateral polarity, acquiring a spindle shape and front–rear polarity; also, the mesenchymal cells acquire the ability to migrate and invade surrounding and distal tissues. Changes in adhesion molecules such as ZO (zonula occludens), occludins, cadherins, mucin, and cytoskeletal proteins such as cytokeratins, occur; also, specific TFs are activated, such as Snail, Twist, and Slug. In addition, epigenetically, each phenotype is regulated by specific miRNAs and IncRNAs. These mesenchymal markers have been associated with tumor progression.

**Figure 2 biomolecules-10-01676-f002:**
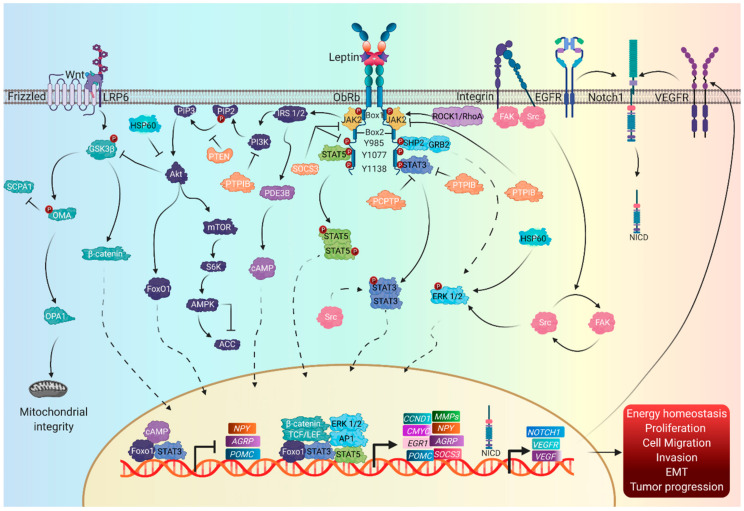
Leptin-induced signaling pathways. The leptin signaling pathway is initiated by leptin–ObR-b receptor interaction, recruitment of ROCK1 (Rho associated coiled-coil containing protein kinase 1), and subsequent activation of JAK2 (Janus kinase 2), which phosphorylates downstream Tyr985, Tyr1077, and Tyr1138 residues. Three canonical signaling pathways that trigger leptin have been reported and are related to the regulation of food intake and energy expenditure. The JAK2-STAT3-5 (signal transducer and activator of transcription 3–5) pathways are the major leptin pathways and induce the expression of appetite–regulating genes and SOCS3 (suppressor of cytokine signaling 3). The MAPK/ERK1/2 pathway is related to the expression of early response genes involved in cell proliferation and differentiation. The IRS1-2-PI3K-Akt pathway is initiated by the direct recruitment of IRS (insulin receptor substrate) to JAK2, triggering the activation of proteins related to the expression of weight-regulating neuropeptides. Inhibition of leptin signaling is mediated by SOCS3 through a negative feedback loop, although PTP1B (protein tyrosine phosphatase 1B) proteins have also been reported.

**Figure 3 biomolecules-10-01676-f003:**
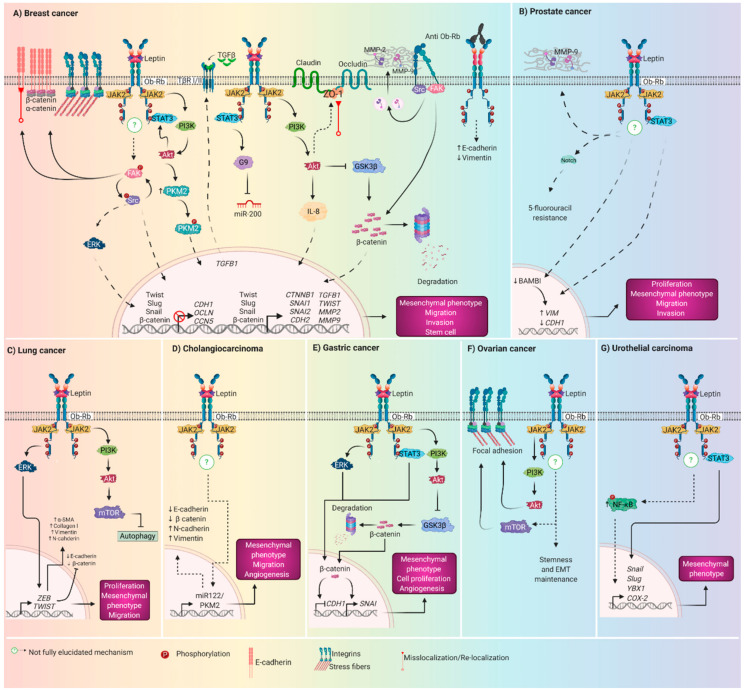
Leptin-induced signaling pathways driving EMT in cancer. Across several signaling pathways, leptin acts as an inducer of EMT in different types of human cancer: (**A**) Breast cancer; (**B**) prostate cancer; (**C**) lung cancer; (**D**) cholangiocarcinoma (**E**) gastric cancer (**F**) ovarian cancer, and (**G**) urothelial carcinoma. Once leptin binds to ObR-b, signaling pathways are activated synergically and converge in the activation or maintenance of EMT through novel and not yet fully described mechanisms (discontinuous arrows) or by activating canonical signaling axis (continuous arrows) such as PI3K-Akt, JAK-STAT-Akt, and Wnt/β-catenin (**A**). Interestingly, in addition to the expected responses to leptin treatment in the different experimental models, other cellular mechanisms seem to actively participate as impairment in cell–cell adhesions (**A**), cytoskeletal remodeling (**A**,**E**), and inhibition of autophagy (**C**). Moreover, stemness characteristics (**F**), chemotherapy resistance (**B**), and angiogenesis (**E**) have been observed.

**Table 1 biomolecules-10-01676-t001:** Epithelial-mesenchymal transition (EMT) inductors.

Inductor		Cell Model	The Effect in the EMT	References
Growth factor	**TGF-β**	HT-29 human colorectal cancer cells	↓ E-cadherin ↓ Cell-Cell contacts ↑ Vimentin ↑ Elongated phenotype	[7]
		SGC-7901 and KATO-III human gastric cancer cells	↓ E-cadherin ↑ Vimentin ↑ N-cadherin ↑ β-catenin ↑ miR-21	[8]
		A549 human lung adenocarcinoma cells H1299 human non-small cell lung carcinoma cells	↑ MMP-2 ↑ MMP-9	[11]
		Endothelial cells	↑ α-SMA ↑ Fibronectin	[10]
		Panc1 human pancreatic cancer cells MDA-MB-231 human breast cancer cells	↑ SNAIL ↑ SLUG	[9]
	**EGF**	AGS, MGLVA1, ST16 human gastric epithelial cells	↑ SNAIL ↑ SLUG ↑ MMP-7	[12]
		HT29 and DLD-1 colorectal adenocarcinoma cells	↓ E-cadherin ↑ SNAIL ↑ ZEB ↑ Vimentin	[13]
		H322, PC9 human non-small cell lung carcinoma CAPAN-2 human pancreatic cancer cells	↓ E-cadherin ↑ Vimentin	[15]
		OVCA429 Human ovarian cancer cells	↓ E-cadherin ↑ Slug ↑ Snail ↑ MMP-2	[16]
		PC3 human prostate cancer cells A549 human lung adenocarcinoma cells	↑ Typical fibroblast-like morphology ↑ N-cadherin ↑ Fibronectin ↑ Snail ↑ Twist	[14]
Cytokines	**IL-6**	Cancer gastric patients’ tissue	↑ α-SMA	[17]
		SGC-7901 human gastric cancer cells	↓ E-cadherin ↑ Vimentin ↑ N-cadherin ↑ ZEB	[17]
		A549 and NCIH358 human lung adenocarcinoma cells	↓ E-cadherin ↑ Vimentin ↑ α-SMA	[18]
	**IL-8**	HCC hepatocellular carcinoma cells	↓ E-cadherin ↑ Spindle-shaped ↑ Snail ↑ N-cadherin	[87]
		Patients with NSCLC	↑ Vimentin ↑ N-cadherin	[88]
	**TNF-α**	SMMC-7721 hepatocellular carcinoma cells	↓ E-cadherin ↑ Spindle-shaped morphology ↑ Vimentin ↑ N-cadherin ↑ β-catenin	[89]
Chemical	**Cadmium**	A549 human lung adenocarcinoma cells	↓ E-cadherin ↑ Spindle-shaped morphology ↑ Formation of stress fibers ↑ N-cadherin ↑ Vimentin ↑ Snail ↑Slug	[19]
		MCF10A nontumorigenic epithelial cells MDA-MB-231, HCC 1937, and HCC 38 human breast cancer cells	↓ Intercellular adhesion ↓ E-cadherin ↑ Spindle-like morphology ↑ N-cadherin ↑ ZEB-1 ↑ Snail	[20]
	**Sodium arsenite, Nickel (II) chloride, and Nickel (II) sulfate**	BEAS-2B human bronchial epithelial cell line	↓ E-cadherin ↑ Fibronectin	[90]
Fatty acids	**Palmitate**	HepG2 human hepatocarcinoma cells	↑ *SNAIL1* ↑ *ZEB2* ↑ *TWIST1*	[91]
	**Free fatty acids**	Patients with primary liver cancer and hepatocellular carcinoma	↑ *SNAIL1* ↑ *VIM* ↑ Wnt and TGF-β signaling	[91]
	**Oleic acid**	MCF10A nontumorigenic epithelial cells MCF7, and MDA-MB-31 human breast cancer cells	↑ MMP-9 ↑ PKC ↑ Src ↑ EGFR	[21]
	**Arachidonic acid**	MCF10A nontumorigenic epithelial cell line	↑ Vimentin ↑ N-cadherin ↑ Cytokeratin 5/8 ↑ MMP-9 secretion	[22]
Oxidative stress and Hypoxia	**ROS**	NRK-52E normal rat kidney cells	↓ E-cadherin ↑ α-SMA	[27]
	**STC2**	SKOV3 human ovarian cancer cells	↓ E-cadherin ↑ Snail ↑ N-cadherin ↑ Vimentin ↑ MMP-2 ↑ MMP-9	[94]
Hormones	**Estradiol and Progesterone**	Embryonic stem cells	↓ E-cadherin ↑ N-cadherin ↑ Snail ↑ Slug	[92]
	**L-thyroxine (T4) 3,5,3′-triiodo-L-thyronine (T3)**	OVCAR-3, SKOV-3 and A2780 human ovarian adenocarcinoma cells	↑ ZEB-1 ↑ β-catenin ↑ Slug ↑ Vimentin	[25]
	**Resistin**	MCF-7, MDA-MB-231 human breast cancer cells MCF10A nontumorigenic epithelial cell line	↑ *SNAI1* ↑ SNAI2 ↑ *TCF8* ↑ *TWIST1* ↑ Snail ↑ Vimentin ↑ Fibronectin ↓ E-cadherin ↓ Claudin-1	[23,24]
Others	**Nicotine**	HK-2 human renal proximal tubular epithelial cells	↑ Vimentin ↑ E-cadherin	[93]

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
