# Peer review of "New Actors Driving the Epithelial–Mesenchymal Transition in Cancer: The Role of Leptin"

_biomolecules, 2020, doi:10.3390/biom10121676_

Round 1
Reviewer 1 Report
The review by Olea-Flores and colleagues is a well-written and -researched manuscript concerning the role of leptin in EMT.
I personally wish there was less of a re-review of EMT and more attention paid to what I perceive their real interest is and that is the influence of leptin in EMT and oncogenesis.
The second half of the article runs out of steam after the section describing the impact of leptin in breast cancer, which seems to be the research interest of the senior author. There appears to be a fair amount of research around leptin influence in other cancer types which would improve the end of the article.
Finally, Figure 3 seems too "busy" in using an individual pane to capture the known influences of leptin in each cancer discussed. Perhaps there is a way to capture these in one figure that describes signaling.
Author Response
Response to Reviewer 1 Comments
Point 1: The review by Olea-Flores and colleagues is a well-written and -researched manuscript concerning the role of leptin in EMT.
Response 1: We appreciate the reviewer's kind comments regarding our manuscript. In relation to the concerns of the reviewer regarding some aspects of our manuscript, we give them an answer point by point below.
Point 2: I personally wish there was less of a re-review of EMT and more attention paid to what I perceive their real interest is and that is the influence of leptin in EMT and oncogenesis.
Response 2: As the reviewer 1 suggested, we decided to summarize the epithelial-mesenchymal transition section and write in depth in the section corresponding to the role of leptin in EMT.
Point 3: The second half of the article runs out of steam after the section describing the impact of leptin in breast cancer, which seems to be the research interest of the senior author. There appears to be a fair amount of research around leptin influence in other cancer types which would improve the end of the article.
Response 3: As suggested by the reviewer, and to complement the section, we decided to add complementary studies that associate leptin with EMT and tumor progression in different types of epithelial cancers such as prostate, renal, ovary, gastric, among others. With these modifications, we think that our manuscript improves the content and quality of this section.
Point 4: Finally, Figure 3 seems too "busy" in using an individual pane to capture the known influences of leptin in each cancer discussed. Perhaps there is a way to capture these in one figure that describes signaling.
Response 4: We appreciate the comments of the reviewer and we decided to summarize the information contained in figure 3. Now, we consider that with the modifications made to the figure, in this new version, figure 3 no longer seems too “busy”. However, we also consider interesting to keeping several subsections to exemplify the molecular events induced by leptin in different types of cancer.
Reviewer 2 Report
Olea-Flores et al. wrote the review article in title, "New actions driving epithelial-mesenchymal transition in cancer: Role of leptin".
On the work, the title is not matched to the content of review manuscript. It is very hard to understand between EMT markers and leptin function. Thus, I will suggest that the review manuscript should divide into two manuscripts (EMT and Leptin). And also, subtitle is too short.
In the manuscript, the authors show the content of Figure 1, BUT not Figure 2 and 3. To make matters worse, the authors do not show the contents following the figures. Thus, I think that the figures are not described by the authors.
The authors show the data derived from a variety of cell lines. However, it is very hard for readers to follow the data of the cell lines. I will do strongly suggest that the authors should make a new table for the cell lines. As too many abbreviations are included in the review manuscript, the authors should show a part of abbreviations for readers.
The authors describe the structure of leptin in the manuscript, but not the figure for it. It is not good for readers.
Collectively, I think the review manuscript do not give us better understanding among EMT, cancer, and leptin.
Author Response
Response to Reviewer 2 Comments
Point 1: On the work, the title is not matched to the content of review manuscript. It is very hard to understand between EMT markers and leptin function. Thus, I will suggest that the review manuscript should divide into two manuscripts (EMT and Leptin). And also, subtitle is too short.
Response 1: We thank reviewer 2 for his concerns regarding our manuscript. First, we would like to clarify that we wanted to make the most representative relationship of the content of the manuscript with the title. In our opinion, the title is clearly related to the content of the present manuscript. Regarding the division of the manuscript in two, we consider that it is not appropriate and we decided to summarize the information of the EMT so that it is more understandable to the end user. On the other hand, we modify the subtitles so that they are not short and are understandable.
Point 2: In the manuscript, the authors show the content of Figure 1, BUT not Figure 2 and 3. To make matters worse, the authors do not show the contents following the figures. Thus, I think that the figures are not described by the authors.
Response 2: Figures 1, 2 and 3 are already cited in the content of the manuscript. Furthermore, we place the citations of the figures according to the content of the figure.
Point 3: The authors show the data derived from a variety of cell lines. However, it is very hard for readers to follow the data of the cell lines. I will do strongly suggest that the authors should make a new table for the cell lines. As too many abbreviations are included in the review manuscript, the authors should show a part of abbreviations for readers.
Response 3: We appreciate the reviewer's suggestion regarding the data of the cell lines; however, we consider that the importance of the cell lines is not in the name, but in the findings obtained. From our perspective, the data presented here are relevant and according to each subsection of the type of cancer, confusion is difficult for the final reader. On the other hand, at the suggestion of the reviewer and the editor, we decided to show the meaning of the abbreviations in the first section where these abbreviations appear for the first time.
Point 4: The authors describe the structure of leptin in the manuscript, but not the figure for it. It is not good for readers.
Response 4: We appreciate the reviewer's comment. However, we consider that describing the receptor structure in a figure is irrelevant for the aim of the present manuscript. There are other reviews published by our working group where we describe the receptor structure.
Point 5: Collectively, I think the review manuscript do not give us better understanding among EMT, cancer, and leptin.
Response 5: We thank the reviewer's opinion. However, to our knowledge, there is no review that examines the role of leptin on EMT in cancer. Here, we report the key role that leptin shows in the activation of the EMT program and consequently of tumor progression in different types of cancer.
Finally, we submitted our corrected version of the manuscript to an extensive English edition and this is the current version with the modifications made to the manuscript.
Round 2
Reviewer 1 Report
My concerns were largely addressed.
On page 2, in line 88, it would seem more appropriate call this "the goals of this review".
In line 93-94 the authors refer to EMT as "reversible" but in line 98 characterize it as "a strict one-way process". These terms seem inconsistent. Also in line 113/114, "MET" is not defined as best I can tell. I believe clarification here would be appropriate.
Otherwise, the manuscript is much improved.
Author Response
Response to Reviewer 1 Comments
Point 1: My concerns were largely addressed.
Response 1: We appreciate the reviewer's kind comments regarding our manuscript.
Point 2: On page 2, in line 88, it would seem more appropriate call this "the goals of this review".
Response 2: This sentence has already been corrected.
Point 3: In line 93-94 the authors refer to EMT as "reversible" but in line 98 characterize it as "a strict one-way process". These terms seem inconsistent.
Response 3: The sentence was changed from “Even though EMT appears to be a strict one-way process from epithelial cells to mesenchymal cells, intermediate states of EMT have been described” by the following sentence: “Even though EMT appears to be a strict complete process from epithelial cells to mesenchymal cells, intermediate states of EMT have been described”.
Point 4: Also, in line 113/114, "MET" is not defined as best I can tell. I believe clarification here would be appropriate.
Response 4: This sentence has already been corrected.
Point 5: Otherwise, the manuscript is much improved.
Response 5: We appreciate the reviewer's kind comments
Reviewer 2 Report
The revised manuscript sounds good compared with the old version. For me, I would like to know the difference or the same between cancer stem cells and EMT progenitor cells.
Author Response
Response to Reviewer 2 Comments
Point 1: The revised manuscript sounds good compared with the old version.
Response 1: We appreciate reviewer 2 for their comments regarding our manuscript.
Point 2: For me, I would like to know the difference or the same between cancer stem cells and EMT progenitor cells.
Response 2: We thank the reviewer for his concern and at the same time the suggestion he makes regarding the difference between cancer stem cells and EMT progenitor cells, however, our interest is not related to those topics. Therefore, these suggestions will not be included in our manuscript.